# Exploring Novel Antidepressants Targeting G Protein-Coupled Receptors and Key Membrane Receptors Based on Molecular Structures

**DOI:** 10.3390/molecules29050964

**Published:** 2024-02-22

**Authors:** Hanbo Yao, Xiaodong Wang, Jiaxin Chi, Haorong Chen, Yilin Liu, Jiayi Yang, Jiaqi Yu, Yongdui Ruan, Xufu Xiang, Jiang Pi, Jun-Fa Xu

**Affiliations:** 1Guangdong Provincial Key Laboratory of Medical Molecular Diagnostics, The First Dongguan Affiliated Hospital, Guangdong Medical University, Dongguan 523808, China; hanbo_yao98@163.com (H.Y.);; 2Institute of Laboratory Medicine, School of Medical Technology, Guangdong Medical University, Dongguan 523808, China; 3The Key Laboratory for Biomedical Photonics of MOE at Wuhan National Laboratory for Optoelectronics—Hubei Bioinformatics and Molecular Imaging Key Laboratory, Systems Biology Theme, Department of Biomedical Engineering, College of Life Science and Technology, Huazhong University of Science and Technology, Wuhan 430074, China; xiang_xufu@hust.edu.cn

**Keywords:** major depressive disorder, novel antidepressants, G protein-coupled receptors, cryo-electron microscopy, virtual drug screening, structure-based drug design

## Abstract

Major Depressive Disorder (MDD) is a complex mental disorder that involves alterations in signal transmission across multiple scales and structural abnormalities. The development of effective antidepressants (ADs) has been hindered by the dominance of monoamine hypothesis, resulting in slow progress. Traditional ADs have undesirable traits like delayed onset of action, limited efficacy, and severe side effects. Recently, two categories of fast-acting antidepressant compounds have surfaced, dissociative anesthetics S-ketamine and its metabolites, as well as psychedelics such as lysergic acid diethylamide (LSD). This has led to structural research and drug development of the receptors that they target. This review provides breakthroughs and achievements in the structure of depression-related receptors and novel ADs based on these. Cryo-electron microscopy (cryo-EM) has enabled researchers to identify the structures of membrane receptors, including the N-methyl-D-aspartate receptor (NMDAR) and the 5-hydroxytryptamine 2A (5-HT_2A_) receptor. These high-resolution structures can be used for the development of novel ADs using virtual drug screening (VDS). Moreover, the unique antidepressant effects of 5-HT_1A_ receptors in various brain regions, and the pivotal roles of the α-amino-3-hydroxy-5-methyl-4-isoxazolepropionic acid receptor (AMPAR) and tyrosine kinase receptor 2 (TrkB) in regulating synaptic plasticity, emphasize their potential as therapeutic targets. Using structural information, a series of highly selective ADs were designed based on the different role of receptors in MDD. These molecules have the favorable characteristics of rapid onset and low adverse drug reactions. This review offers researchers guidance and a methodological framework for the structure-based design of ADs.

## 1. Introduction

Major Depressive Disorder (MDD), a severe mental illness, exhibits a global lifetime prevalence of approximately 15% [1]. Nevertheless, the diagnosis and treatment of depression face challenges. The lack of well-defined pathological mechanisms and drug targets often leads to traditional antidepressants (ADs) focusing on the monoamine system. As a result, they frequently have severe side effects due to their non-specific binding to multiple amine receptors [2,3]. These medications display an extended onset of action, taking several weeks [4,5,6]. Additionally, one-third of patients experience inefficacy and a vulnerability to relapse, a condition termed “treatment-resistant depression (TRD)” [7]. Hence, the identification of the core nodes governing depression and the development of effective compounds have emerged as primary goals [3,8,9].

Since the 1990s, advancements in high-throughput screening methods for cellular and biochemical assays have expedited the discovery of high-affinity molecules [10]. However, these compounds cannot achieve selective activation of specific receptor subtypes and downstream signaling pathways [11]. Consequently, candidate molecules are frequently excluded from rigorous clinical trials due to unforeseen adverse effects [12]. Since the human G protein-coupled receptor (GPCR) was first resolved by X-ray crystallography in 2007, a surge of targeted molecule discovery methods based on structural information has ensued [13]. Furthermore, advancements in cryo-electron microscopy (cryo-EM) have empowered researchers to acquire receptor structures in their activated state [14,15]. In the meantime, the rapid expansion of drug libraries has enabled researchers to rapidly design or screen desired molecules from the vast chemical space, a theoretical space that encompasses all possible chemical substances, based on known ligand binding fingerprints and molecular interactions [16,17].

A groundbreaking drug, S-ketamine, has received Food and Drug Administration (FDA) approval for the treatment of patients with TRD and suicidal tendencies [18]. This signifies the successful development of the first AD that is not based on the monoamine hypothesis in 50 years. Similarly, psychedelics, such as psilocybin, have shown effective antidepressant properties in clinical studies by activating 5-hydroxytryptamine (5-HT) receptors [19,20]. These compounds have garnered significant attention. Although these compounds can have rapid and effective antidepressant effects at low doses, their inherent addictive and hallucinogenic properties pose challenges for their clinical endorsement [21,22,23]. Cryo-EM has provided the atomic coordinates of the pockets between novel ADs and their main interacting receptors, the 5-HT_2A_ receptor and the NMDAR [24,25,26]. Leveraging these high-resolution structural insights, a series of pure, low-side-effect, directionally, and highly selectively activating molecules have been designed [27].

In this review, the concrete manifestations of cross-scale abnormalities that contribute to the unclear mechanisms of MDD are initially illustrated. Subsequently, we introduce the advances in cryo-EM technology and VDS. Emphasis is placed on the explosive expansion of virtual drug libraries and the application of artificial intelligence (AI) to structural prediction. Finally, a comprehensive summary is presented, encompassing the mechanisms and structural knowledge of receptors, as well as potential antidepressant compounds. The drugs binding fingerprints of key receptors, including the N-methyl-D-aspartate receptor (NMDAR), tyrosine kinase receptor 2 (TrkB) in the glutamate system, the 5-HT_2A_ receptor, the 5-HT_1A_ receptor, and nitric oxide synthase (nNOS) in the monoamine system, are summarized. The common and unique characteristics of drug development approaches in utilizing structural information are discussed. It should be noted that this review does not intend to completely summarize the mechanisms of all novel ADs but rather to indicate the direction for the discovery of a new generation of structure-based ADs.

## 2. Challenges in MDD: Cross-Scale Abnormalities

MDD is widely recognized as a multifaceted syndrome and symptomatology, including feelings of guilt, hopelessness, psychiatric and cognitive impairments, disturbances in sleep and appetite, and various other manifestations [1]. Given the absence of diagnostic biomarkers, distinguishing between heterogeneous patients with distinct pathophysiological mechanisms poses a significant challenge [7]. Furthermore, these symptoms often overlap with the diagnosis of other psychiatric disorders, resulting in a crude and difficult formulation of treatment plans [28]. The hypothesis for depression crosses multiple systems, brain regions, and neurons [29,30,31]. The disruption of neural connectivity networks, widespread structural changes in the brain, and abnormal neurotransmitter transmission have been widely recognized as playing an important role in the onset of MDD [32]. With the remarkable advancements in biological imaging and microscopy techniques, the scope of depression research has transcended from the macro- to the microscale and even delved into the nanoscale [15,33].

At the scale of the endocrine system, the dysregulation of the hypothalamic–pituitary–adrenal axis is a longstanding area of investigation. Stress induces excessive cortisol release and disrupts feedback mechanisms, leading to a substantial rise in plasma cortisol levels among patients [34]. However, the administration of glucocorticoid receptor antagonists fails to elicit antidepressant effects in clinical settings [35,36]. Another aspect, the inflammatory hypothesis, posits that the aberrant stimulation of the nervous system by the immune system serves as a significant etiological factor of MDD. This perspective is based on the increased inflammatory markers in patients, including Interleukin 6 (IL-6) and C-reactive protein (CRP) [37,38]. Peripheral cytokines traverse the blood–brain barrier, exerting their influence on neurons and support cells, thereby contributing to deleterious alterations in brain structure and functionality [39,40]. Despite significant research efforts, establishing a direct and conclusive link between these abnormalities and MDD remains a challenging task. Furthermore, there is a shortage of ADs that are supported by substantial evidence and that effectively target these abnormalities.

By means of autopsy and magnetic resonance imaging (MRI) of patients, abnormalities in the cerebral regions that are associated with MDD have been shown to manifest in the cortex and subcortex, including the hippocampus (HPC), amygdala (Amyg), nucleus accumbens (NAc), and medial prefrontal cortex (mPFC) [41]. The atrophy of the HPC volume is highly correlated with the course of depression, and transcranial stimulation of the mPFC alleviates depressive symptoms [42,43]. In addition, the negative bias-related memories that are associated with depression are linked to the HPC, Amyg, anterior cingulate cortex (ACC), mPFC, and NAc [44]. MRI has revealed atrophy in the mPFC and ACC of patients [42]. The aberrations in the reward circuit are also key factors in triggering MDD. In the reward circuit, NAc exerts a pivotal function in mediating emotional dysregulation behavior by integrating excitatory glutamatergic neurons from the HPC, hypothalamus (HT), and mPFC (Figure 1, process 1) [30,45]. Insufficient excitatory projections to the NAc lead to reduced excitation and brain volume [46]. And the feedback loop between mPFC-NAc-HT is also necessary to maintain brain reward states [30]. The ventral tegmental area (VTA)-NAc’s dopaminergic circuit has significant antidepressant properties after optogenetic activation (Figure 1, process 2) [47].

The lateral habenula (LHb) occupies a unique position in depression, closely interacting with all midbrain neuromodulatory systems, including the noradrenergic, serotonergic, and dopaminergic systems [48,49,50]. The LHb can activate the rostromedial tegmental nucleus (RMTg), which in turn can inhibit VTA neurons, ultimately leading to the emergence of negative emotions (Figure 1, process 3) [48,51,52]. Studies have shown that the optogenetic activation of LHb input induces strong avoidance behavior in mice [53]. Hence, depleting LHb neurotransmitters via deep brain stimulation to cancel the inhibition of the VTA can reverse depression-like behavior [54]. Further studies have found that LHb produces specific cluster discharges in depression animal models, strongly inhibiting reward-related DRN and VTA (Figure 1, process 4) [55,56]. This discharge anomaly can be improved by ketamine [57,58]. In addition, the development of new animal models has shown that LHb does activate in response to psychosocial stress, which then leads to depressive responses. In a mouse model of forced defeat due to low social status, reward prediction errors strongly activated LHb and inhibited mPFC [59].

Although MDD involves abnormalities in multiple brain areas, there are two common patterns at the neuronal level: neuronal atrophy and discharge abnormalities. The mechanism of neuronal atrophy can be explained by the synaptic plasticity hypothesis, which suggests that the efficacy and size of dendritic spines vary significantly in response to stimulation [60,61]. Hebbian plasticity is a positive feedback mechanism that operates on a scale of seconds to minutes. When the axon repeatedly and continuously emits signals, the dendritic spine responds to the same stimulus by increasing its efficiency, resulting in long-term potentiation (LTP) [62,63]. In a healthy condition, the activation of TrkB promotes the influx of Ca^2+^ and the expression of postsynaptic density (PSD) proteins, including Calcium-/calmodulin-dependent protein kinase type II (CaMKⅡ), postsynaptic density protein 95 (PSD-95), and the α-amino-3-hydroxy-5-methyl-4-isoxazolepropionic acid receptor (AMPAR). This process leads to the thickening of PSD and the enlargement of dendritic spines [64,65,66]. In a depressive condition, persistent stress diminishes the brain-derived neurotrophic factor (BDNF) levels and impairs LTP, leading to an imbalance in plasticity, inducing synaptic dysfunction, and resulting in signal loss [67,68]. The loss of synaptic plasticity is associated with the weakening of TrkB signaling. Traditional ADs, ketamine, and psychedelics can all activate synaptic plasticity and synaptogenesis [69,70,71]. This means that synaptic plasticity is located at the hub for rapid antidepressant effects and explains disorders at the subcellular and protein levels.

Abnormal discharges are manifested at the protein level as ion channel-mediated abnormal ion influx. Since blocking 5-HT reuptake can alleviate depressive symptoms, the traditional view is that MDD is associated with a loss of 5-HT receptor signaling [72,73]. Normally, activated 5-HT receptors activate downstream kinases, ion channels, and signal transporters. This leads to Ca^2+^ release from the endoplasmic reticulum and the opening of postsynaptic Ca^2+^ channels, thereby transmitting excitatory signals [74,75]. However, since ketamine, as an ion channel antagonist, has effective antidepressant effects, the focus of research on the mechanism of MDD has gradually shifted to the NMDAR [58,76,77]. The activation of Ca^2+^ channels in inappropriate brain regions and subcellular locations (such as LHb and extrasynaptic sites) leads to pathological conditions [58,78]. The direct binding of psychedelics to 5-HT receptors and ketamine to NMDARs prompted structural biologists to tap into the mechanism of action of ADs at the nanometer and atomic scales. This includes the biased activation of the 5-HT_2A_ receptor by psychedelics and the NMDAR antagonism of ketamine [25,26,79].

In summary, technological advancements will facilitate the exploration of mechanisms underlying depression at the subcellular and even protein levels, transitioning from the macroscopic to the microscopic scale. Due to the involvement of multiple brain regions and neurons, as well as multiple membrane receptors and downstream signaling proteins in the 5-HT and glutamate systems, it is a challenge to design high-efficient and low-side-effect ADs. 

## 3. Controversial ADs: Psychedelics and Ketamine

### 3.1. Psychedelics: 5-HT_2A_ Receptor Agonists

In the mid-20th century, classic psychedelics were used in psychedelic therapy and psychotherapy to treat a variety of mental illnesses, including depression, anxiety, and personality disorders [80]. Despite producing significant therapeutic effects, their strong psychedelic experiences, such as visual distortions, falling illusions, and “oceanic” states of consciousness led to their abuse during the hippie movement of the 1960s and subsequent regulatory control by multiple countries [22,81,82]. The most famous regulation is the United States’ Federal Controlled Substances Act (CSA). This led to a slump in psychedelic research for 50 years. However, in the 1980s, psychedelics were confirmed to activate the 5-HT_2A_ receptor and exert an emotional regulation function [83,84]. In 2016, when the enduring antidepressant effects of psychedelics were substantiated, this rekindled the enthusiasm for further exploration of their therapeutic effects [19,85].

Psychedelics can be classified into two categories according to their sources: natural products such as psilocybin, N,N-Dimethyltryptamine (DMT), and mescaline and synthetic and semi-synthetic drugs, represented by ergotamine derivatives [86,87,88]. They can also be divided into three types according to their chemical structure: indoleamines, phenylalkylamines, and semi-synthetic ergotamines [89]. Phenylalkylamines such as 2,5-Dimethoxy-4-iodoamphetamine (DOI) and 4-Bromo-2,5-dimethoxyamphetamine (DOB) are selective agonists of 5-HT_2_ receptors, with an affinity for 5-HT_2_ receptors that is 100 to 1000 times that of 5-HT_1/5_ receptors (Figure 2A) [90,91]. Indoleamines (such as psilocin and psilocybin) and ergotamines, such as lysergic acid diethylamide (LSD), exhibit extensive agonistic interactions with serotonin receptors, including 5-HT1/2/5/6 and 7 [92,93,94]. LSD, as a representative of semi-synthetic psychedelics, has a very high psychedelic effect and acts on other GPCR families [95,96]. The evolutionary relationships of the 5-HT receptor family and the corresponding highly affinity G-protein members are summarized below (Figure 2B).

Because of their tetracyclic structure, ergolines have the broadest range of 5-HT family receptor binding capacity. Thus, there are 12 ergoline-binding structures, and the 5-HT_2B_ receptor, an alternative to the 5-HT_2A_ receptor, was most comprehensively resolved and obtained in multiple states of transducer-free, binding Gq and β-arrestin (Table 1) [97]. Psilocin showed antidepressant effects in indoleamines in clinical trials, and thus, the psilocin-5-HT_2A/C_ receptor was acquired [25,98]. Of the NBOMe series, 25CN-NBOH is reported to be among the most potent and selective in vitro and in vivo. Therefore, the 25CN-NBOH-5HT_2A_ receptor was compared with the partial agonist LSD to understand the structural mechanisms of its activation strength [24].

Although psychedelics activate a wide range of 5-HT receptors, the hallucinogenic effects are believed to arise from the activation of the 5-HT_2A_ receptors [99]. Psychedelics activate 5-HT_2A_ receptors on cortical and subcortical brain areas, particularly on layer 5 pyramidal neurons [89,100]. Additionally, there is a perspective suggesting that psychedelics demonstrate a common β-arrestin bias, which plays a role in the psychedelic experience [100,101]. Recently, it has been proposed that hallucinogenic effects result from the simultaneous activation of G proteins and β-arrestin, and that designing biased partial agonists could potentially reduce psychedelic responses [25].

However, indoleamines and ergotamines also exhibit a high affinity for the 5-HT_1A_ receptor. The hallucinogenic and antidepressant effects are thought to originate from the 5-HT_2A_ receptor, and the contribution of the 5-HT_1A_ receptor is mentioned less frequently [102,103,104]. In addition, the off-target effects of psychedelics on the 5-HT_2B_ receptor and the 5-HT_2c_ receptor can, respectively, induce lethal heart valve and anorexia [105,106,107]. And co-administration with other drugs targeting the serotonergic system can trigger serotonin syndrome [108]. This symptom leads to abnormalities in the nervous system, autonomic nervous function, and neuromuscular innervation, and can even be fatal.

**Table 1 molecules-29-00964-t001:** Key information on different classes of psychedelics.

Class	Representation	Compounds	Binding Structure (PDB ID)	Clinical Trials
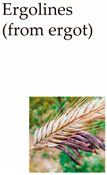	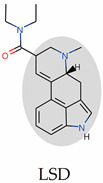	LSD, ergotamine (ERG), dihydroergotamine (DHE)	5-HT1BR-ERG (7C61) 5-HT1BR-DHE (4IAQ) 5-HT2AR-LSD (7WC6) 5-HT2BR-LSD (7SRS) 5-HT2BR-methysergide (6DRZ) 5-HT2BR-methylergonovine (6DRY) 5-HT2BR-ERG (5TUD) 5-HT2BR-LSD (5TVN) 5-HT2BR-ERG (4NC3) 5-HT2CR-ERG (6BQG) 5-HT5AR-methylergonovine (7UM7)	a. LSD-assisted therapy with anxiety and ratings of depression symptoms [109]. b. Single microdoses of orderly produced LSD, dose-related subjective effects [110]. c. The link between psychosis model and therapeutic model seems to lie in LSD mystical experiences [111].
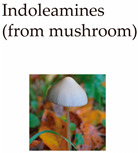	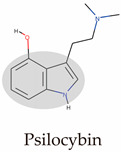	DMT, 5-MeO-DMT, psilocin, psilocybin	5-HT2AR-psilocin (7WC5) 5-HT2CR-psilocin (8DPG)	a. Compared trial: psilocybin versus escitalopram for depression [112]. b. Assisted therapy: psilocybin was given in the context of supportive psychotherapy [113]. c. Psilocybin for TRD [114]. d. After psilocybin therapy for depression, global integration in the brain is increased [115].
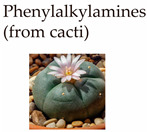	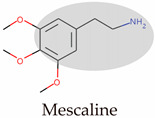	Mescaline, DOM, DOI, DOB, NBOMes	5-HT2AR-25CN-NBOH (6WHA)	(none)

Psychedelics have multiple pharmacological effects on 5-HT receptors: system bias, ligand bias, and receptor bias; these results in diverse physiological effects [116]. In recent years, there has been investigation into the specific mechanisms of hallucinogenic effects to separate antidepressant effects from hallucinogenic effects [25,117].

### 3.2. Ketamine: An Antagonist of the NMDAR

The need for anesthetics during wars in the last century prompted the synthesis of ketamine. It offered the advantage of a rapid onset of action compared to its prototype, promazine [118]. However, ketamine has serious side effects, and overdosing can lead to respiratory arrest and even death, accompanied by mental poisoning and symptoms of schizophrenia [119]. Its excitatory effects on the sympathetic nervous system can cause tachycardia and hypertension, and its activation of opioid receptors leads to “dissociative sensations”, a state of separation of consciousness and sensation [120,121,122]. Therefore, it has been classified as a controlled substance by several countries.

It was discovered in the last century that subanesthetic doses of ketamine have the potential to relieve psychiatric disorders and depression [123,124]. However, the lack of clear effects on the monoamine system, which was the mainstream hypothesis for depression at the time, limited its further development. And it was shelved with the promulgation of the ban. Clinical studies in the early 21st century reawakened the use of ketamine [76]. Under double-blind and placebo-controlled conditions, ketamine can reduce depression scores within four hours and has a lasting effect of three days, while its harmful effects disappear rapidly after infusion. This means that ketamine has significant clinical value, which has spurred multiple clinical studies on TRD, with results showing that ketamine has rapid and long-lasting antidepressant effects [125,126,127].

Ketamine has been identified as a non-competitive NMDAR antagonist [128]. As a calcium-permeable receptor of ion channels, the NMDAR has a calcium permeability that is four times that of the same type of receptor, the AMPAR [129]. The high sensitivity and slow desensitization of NMDAR glutamate make it the main carrier of Ca^2+^ transport. Ketamine, when used as an anesthetic in clinical practice, is a racemic mixture of S-ketamine and R-ketamine. Although only S-ketamine is approved as a prescription drug, many clinical studies have shown that R-ketamine also has a rapid onset of antidepressant action [130,131]. After ketamine enters the body, it is further metabolized into 2R,6R-hydroxynorketamine (R-HNK) [132], which does not bind to the NMDAR, and its antidepressant effect is mediated by synaptic plasticity from AMPAR and TrkB activation [133,134].

## 4. Advancements in Cryo-EM and VDS 

### 4.1. Cryo-EM: Resolving Active Receptors

Although over 85% of protein structures in databases are provided by X-ray [116], the resolution of cryo-EM has surpassed 5 Å since 2013 following advancements in electron detectors and image analysis techniques [117,135,136,137,138]. This has provided an advantage in analyzing highly dynamic membrane proteins and large complexes. This stems from several advantages of the imaging method: (1) it does not require crystallization, making it not limited by molecular weight; (2) it allows for direct imaging in artificial membrane mimics, which aids in resolving membrane proteins; and (3) three-dimensional classification algorithms can elucidate highly dynamic active protein structures [139,140,141]. In drug discovery, cryo-EM has advanced the imaging of membrane proteins, particularly GPCRs. Following the publication of the first cryo-EM structure of the calcitonin receptor coupled with Gs in 2017, there has been a growth in the availability of accessible GPCR structures [140]. Cryo-EM has a lower threshold for resolving active GPCR structures than X-ray, and a lower protein quality and purity can achieve a higher resolution. The development of agonists is more attractive than antagonists, and the conformation of agonist-bound GPCRs that are resolved by X-ray is in an intermediate state between active and inactive [142,143,144]. Since 2017, 811 GPCR structures have been published, and cryo-EM has obtained 472 active structures (Figure 2C) [145]. The mechanisms of and advancements in cryo-EM technology have been thoroughly summarized [15].

### 4.2. Molecular Docking and Virtual Drug Libraries

Although the concept of molecular docking and structure-based drug design was proposed in the late 19th century and the 1970s, it was not widely used for drug screening until the past decade. This was because the first human GPCR structure was only solved in 2007, and the first 5-HT_2_ receptors’ family structure was solved in 2013 [143,146]. Prior to this, GPCR structures were modeled based on rhodopsin and were not accurate [147]. In addition, the available chemical space was relatively limited, with the initial version of the ZINC database in 2005 containing only 5 million molecules. Today, the number of molecules that can be directly used has reached billions [148,149]. Molecular docking technology aims to predict ligand-target protein binding modes and affinities through computer calculations, replacing traditional screening methods such as cell assays (e.g., bioluminescence resonance energy transfer, cell viability, reporter genes, microscopy screening) or biochemical assays (e.g., fluorescence resonance energy transfer, surface plasmon resonance, nuclear magnetic resonance) [150]. This will save significant time and costs and increase the number of screening objects by several orders of magnitude [151].

The main steps of molecular docking include sampling and scoring. First, a large number of possible binding modes are generated by docking, and then, they are scored using a physical and experimentally calibrated scoring function to obtain the most likely result [152]. The most popular docking programs that are used are the open-source program DOCK, developed in the 1980s, and the commercial program Schrodinger’s Glide [153,154]. In comparative studies of multiple programs, different programs have advantages for different types of proteins [155]. In addition, accurate model construction, molecular dynamics simulations (MDSs) to obtain multiple conformational energies, and wet experiments providing binding sites effectively enhance docking accuracy [156,157]. This assumes the limitations of molecular docking. In cases where the structure and reference binding modes are lacking, the results can only offer limited guidance.

Virtual screening is the large-scale application of molecular docking, which involves docking millions or even billions of molecules with target proteins and selecting high-scoring molecules for synthesis and validation. The general steps include (1) the design of chemical libraries, (2) the selection and preparation of receptor structures, (3) the evaluation of docking performance, (4) docking screening and compound selection, (5) experimental evaluation, and (6) hit-to-lead optimization [9]. The design of chemical libraries and the selection and preparation of receptor structures have made significant progress in recent years and are explained here. The precautions for the remaining parts and the practical guidelines for large-scale virtual screening have been summarized [9].

The most significant advancement in chemistry libraries in recent years is the increased number of molecules. The version update of the ZINC database boosts the number of compounds from millions to billions. [149,158]. Most of the added molecules come from on-demand synthesis: these molecules have not yet been synthesized, but they can be easily synthesized using a large number of optimized reactions and composition modules. As the largest REadily AccessibLe (REAL) drug library supplier in ZINC, Enamine’s drug space can produce more than 11 billion drug-like compounds through on-demand synthesis.

A study generated 170 million custom molecules from 130 well-characterized reactions in Enamine’s REAL library and docked them against targets [16]. The expansion of the docking library was found to effectively increase hit rates, with an astonishing 453,000 ligands docked successfully. Of the newly discovered chemical types, 30 molecules showed submicromolar activity, which means that expanding the chemical space through custom synthesis will help discover high-affinity molecules with novel scaffolds. Further screening results were obtained by comparing the results of docking a massive database of 1 billion virtual molecules with those of a stock database of 3.5 million synthesized molecules [17]. This showed that the results of docking a super large database no longer favored bio-like molecules, which were reduced by 19,000 times compared to the stock database. This suggests that expanding the screening space is advantageous for discovering ligands with new chemical types. 

To improve the efficiency of large-scale virtual screening, the VirtualFlow virtual screening platform can scale linearly with a higher CPU number, allowing for the docking of a billion compounds to be shortened to two weeks when 10,000 CPUs are used [159]. Another study used a modular collaborative evolution method, V-SYNTHEs, for the partitioned structural screening [160]. Initially, flat compounds that represent all compounds were pre-docked to determine the best scaffold–synthetic compound combination, and then, iterative elaboration was performed to select the best molecules. Predicting the best compounds requires docking of less than 1.0% of members. This method increased the speed and success rate by 5000 times compared to complete screening.

Based on the favorable results of virtual screening of a massive database, selective agonists for the melatonin receptor 1 (MT1), rather than the MT2 receptor, were discovered from 150 million drug-like compounds in ZINC15 [161]. The newly obtained agonist advanced the phase of the mouse circadian clock by about 1.5 h. Another study determined the structure of the sigma-2 receptor, resident in the endoplasmic reticulum, and obtained highly selective molecules that activate the sigma-2 receptor rather than the sigma-1 receptor from 490 million molecules [162]. The molecule caused a decrease in mechanical hypersensitivity in a neuropathic pain model, confirming the unique role of the sigma-2 receptor in pain, independently of the sigma-1 receptor. This indicates that large-scale virtual screening can help obtain chemical probes to confirm unclear receptor functions.

In addition to discovering subtype-selective ligands, designing low-side-effect small molecules holds further interest. This requires precise control of the activation strength and downstream pathway selection [163]. A super large virtual screening for non-sedative analgesic molecules targeting the alpha-2A adrenergic receptor (a2AAR) obtained 20 million fragment compounds and 280 million drug-like compounds [164]. The results showed that fragment molecules accounted for 90% of the final results, which means that for target proteins with smaller binding pockets, fragment molecules are more likely to have a high affinity than drug-like molecules that are commonly used for docking. The obtained agonists preferentially activated Gi, Go, and Gz subtypes, rather than G proteins, and β-arrestin was widely activated by traditional analgesics. Cryo-EM was used to obtain the structure of a newly coupled molecule of the a2AAR, leading to the discovery of a more potent compound—PS75. In animal experiments, these agonists effectively alleviated pain behavior in various pain experiments, even at high doses, without causing sedation. Another screening, conducted during the same period, utilized a rare chemical scaffold to target 5-HT_2A_ receptors and designed non-hallucinogenic antidepressant molecules, which are described in detail below [165].

### 4.3. Predicting Structures via Artificial Intelligence

In the early 21st century, molecular docking was conducted to discover selective antagonists within the 5-HT_2_ receptor family [166]. This study was based on homology modeling using the rhodopsin receptor. Even earlier, 5-HT_1A_ receptor agonists were designed based on basic charge characteristics [167]. However, due to the unavailability of the real structure, the designed molecules often exhibited broad binding. In subsequent optimizations, multiple tests were required to analyze pharmacophore functionality, resulting in significant costs [168,169].

The need for unknown target protein–ligand binding structures led to the organization of the Critical Assessment of Structure Prediction (CASP) competition [170]. The organizers provided the names of the unreleased target protein and their corresponding ligands and released the actual structure after the participants completed the modeling, scoring the accuracy of the prediction. Before 2018, most teams had an accuracy of less than 50%. However, the inclusion of the artificial intelligence (AI) programs AlphaFold V1.0 and AlphaFold 2 led the DeepMind team to win the championship, achieving accuracy rates of 70% and over 90%, respectively, due to the advantages of neural network algorithms over traditional homology modeling [171,172]. First, the database is more comprehensive, including multi-species macro-genomic data and all the structures that are included in the Protein Data Bank (PDB), with multiple rounds of iterative optimization. Then, AlphaFold can extract co-evolutionary information from multiple sequence alignments, identifying residue pairs that are distant in sequence but interact in three-dimensional space. This capability is particularly valuable for GPCRs that lack homologous templates [172,173]. Moreover, homology modeling necessitates redocking multiple obtained structures with known ligands to select the model with the highest affinity, which can introduce structural bias and reduce repeatability [174].

However, there are still limitations in AI structure prediction. Reproducing known ligand–GPCR structures using predictive models revealed that AlphaFold demonstrates high accuracy for the main chain but relatively low accuracy for the side chains [175]. This stems from the reshaping of flexible residues in the pocket by the ligand and the mixture of different states of GPCRs in the database. For GPCRs with extensive extracellular structures, accurate fixed structures were established for both the extracellular and transmembrane domains. However, relative positioning displacement occurred because of uncertainties at the junctions. This is because the disordered sequences at the junctions are often highly dynamic and can be stabilized through the construction of GPCR dimers or multimers and the use of nano-antibodies in experiments. Additionally, comparing transmembrane helix 6 (TM6), TM7, and key motifs revealed that Alphafold tends to overlap active and inactive states, limiting its use for drug screening.

To enable the application of Alphafold for the directed prediction of multiple conformations of GPCRs under physiological states, various modifications have been attempted in the Alphafold process, including (1) selecting a small subset of sequences for a more shallow multiple sequence alignment [176]; (2) masking or mutating segments of the sequences that may introduce bias toward specific conformations [177]; and (3) supplying Alphafold with homologous templates from annotated active state databases [178]. In addition to algorithm modifications, MDSs are used as auxiliary methods to provide insights into the specific physiological states of intermediate proteins [179,180]. MDSs can operate at atomic resolution over timescales ranging from microseconds to milliseconds. While these methods effectively yield active and inactive conformations, the direct generation of conformations under varying signal strengths and with distinct molecular partners remains a challenge.

Another important issue is whether AI can be used for the construction of protein complexes or for predicting protein–protein interactions. By combining the monomer structures that were predicted by Alphafold with cryo-EM maps, the nuclear pore was reconstructed at a high resolution from an intermediate resolution, and Alphafold effectively identified the key interactions between subunits [181]. Another study used multiple sequence alignments of 8.3 million pairs of yeast proteins to screen for 1505 pairs of potentially interacting proteins [182]. Alphafold was iteratively optimized by the Multimer algorithm to predict complex structures, which showed higher accuracy than docking [183]. Similarly, Alphafold-Complex completed the construction of large complexes without the need for pairwise sequence alignment [184]. Can artificial intelligence quickly pair and assemble members through simple inputs of GPCRs and transporters and provide binding pockets for drug discovery with functional selectivity? Or can it resolve the GPCR–G protein-coupled receptor kinase (GRK) rapid dynamic enzyme reaction system [185]? These will be the main topics of the next stage [27,186].

Although the first structure-based prescription drug has yet to emerge, considering three mainstream drug targets reveals challenges: (1) kinases, with their dynamic activation loop and phosphorylation process, hinder the identification of specific fingerprints; (2) GPCRs were not characterized in their active state until 2017; and (3) ion channels, often forming multimeric complexes, present difficulties in terms of structural utilization. More importantly, the process from discovering potential molecules to ultimately launching them on the market takes 10–20 years of animal and clinical trials [187]. Nonetheless, there are compelling reasons to anticipate that, in the foreseeable future, numerous first-in-class drugs, unearthed through the integration of cryo-EM and deep learning, will emerge. These compounds are expected to offer advantages in terms of safety and efficacy that are currently absent in traditional drugs. Especially in the field of psychedelic therapy for depression, this ideal is being realized. Here, we summarize widely used and recently developed virtual screening tools (Table 2).

## 5. Non-Hallucinogenic Psychedelics

### 5.1. Functionally Directed Approach and Fluorescence Sensors

The two non-hallucinogenic psychedelic analogs, tabernanthalog (TBG) and AAZ-A-154, were not designed based on structural information [196,197]. The prototype of TBG is the non-classical psychedelic ibogaine, which possesses the ability to counteract addiction and depression through synaptic plasticity [198,199]. However, its strong inhibition of hERG potassium ion channels can cause cardiac toxicity, even leading to death [200,201]. A functionally directed approach was used to simplify ibogaine and obtain TBG by removing the quinoline ring, which significantly reduces its lipophilicity and prevents binding to hERG channels. TBG has lower hallucinogenic effects, induces synaptic plasticity, and exhibits antidepressant and anti-addictive properties. TBG selectively binds to the 5-HT_2_ receptors and acts as an agonist of the 5-HT_2A_ receptor and an antagonist of the 5-HT_2B_ receptor, avoiding heart valve damage caused by 5-HT_2B_ receptor activation [105].

AAZ-A-154 was designed during the development of fluorescence sensors [197,202] to address the issue of the limited in vitro detection methods for psychedelic drugs and bridge the gap between cell-based assays and human behavioral studies. The chemical probe psychLight was synthesized, which replaced the intracellular loop 3 (ICL3) sequence of the 5-HT_2A_ receptor with the circularly permuted green fluorescent protein (cpGFP) to emit a fluorescent signal upon allosteric activation [203]. This probe was used to screen molecules that compete with 5-HT but only activate the 5-HT_2A_ receptor to a low level, thus avoiding hallucinogenic effects. The 5-methoxyindole derivative AAZ-A-154 was obtained, which exhibits rapid and long-lasting antidepressant effects after a single dose. 

Subsequent studies by the same group confirmed that psychedelics produce antidepressant effects by specifically activating intracellular subgroups of the 5-HT_2A_ receptor [71]. This highlights how cellular sublocalization bias can influence the diverse physiological effects that are induced by various ligands. A contradiction exists, as 5-HT, a full agonist of G proteins and β-arrestin, has no hallucinogenic effect [204]. This is attributed to the subcellular localization bias of 5-HT and psychedelics, which leads to the activation of receptors in distinct subcellular sites. As a polar molecule, 5-HT relies on the serotonin transporter (SERT) for transport into the cell, while the N-methylation modification of psychedelics enables them to enter more easily [71,205]. The 5-HT_2A_ receptor is enriched on the Golgi apparatus, where there is a slightly acidic microenvironment, allowing psychedelics to bind and protonate to maintain their long-term efficacy [206]. By modifying the structure of 5-HT or overexpressing SERT in the cortex to increase the inward transport of 5-HT, it was found that 5-HT leads to similar activation of synaptic plasticity and the head-twitch response (HTR) as psychedelics.

Therefore, the inherent perspective that psychedelics exert their main functions in the plasma membrane is subverted. Producing synaptic plasticity effects must depend on the activation of the intracellular 5-HT_2A_ receptor, although its downstream association with synaptic plasticity pathways is still unclear. This explains why blocking the 5-HT_2A_ receptor on the plasma membrane does not affect the antidepressant effect of psilocybin [207]. Interestingly, this conflicts with the traditional monoamine hypothesis, as the overexpression of SERT triggers rapid antidepressant effects of 5-HT in this research. Therefore, it is necessary to reexamine the contributions of different signaling pathways that are activated by ADs to determine whether exciting 5-HTergic neurons or inducing synaptic plasticity is the key mechanism of antidepressant efficacy [208,209,210].

While functionally directed molecular design and fluorescence sensors have led to the discovery of non-hallucinogenic antidepressant compounds, the number of candidate molecules remains relatively limited, comprising only 14 and 83 compounds, respectively. Furthermore, the synthesized compounds still bear a resemblance to psychedelic drugs and are constructed based on a simplified model of GPCR activation and deactivation. The 5-HT_2A_ receptor-binding psychedelic structures provide insights that subsequently guide the development of antidepressant molecules using VDS.

### 5.2. Structures of the 5-HT_2A_ Receptor

Due to the low expression of the 5-HT_2A_ receptor during protein synthesis, the highly homologous 5-HT_2B_ receptor has been used as an alternative model to study the effects of psychedelics. The first members of the 5-HT receptor family to be determined were the 5-HT_1B_ and 5-HT_2B_ receptors, which bind ergotamine (LSD-like Parkinson’s drug) [143,144]. The first LSD-binding structure was subsequently released [211]. The simpler secondary pharmacophore LSD allows the extracellular loop 2 to form a “lid” that closes the drug-binding pocket and inhibits the slow dissociation of LSD, leading to biased activation of β-arrestin. This also explains the persistent effects of LSD.

The structures of 5-HT_2A_ receptor-binding LSD (partial agonist), 25-CN-NBOH (high-potency phenethylamine agonist), and methiothepin (inverse agonist) were released at the same time [24]. All three ligands interact with W6.48 (Ballesteros–Weinstein numbering) [212]. For class A GPCRs, the Ballesteros–Weinstein numbering scheme means that residues in a TM helix (X) are numbered relative to the most conserved amino acid, which is defined as X.50. The most critical one is the conformation of the 25-CN-NBOH-5-HT_2A_R-Gq active complex (Figure 3A). Alignment showed that 25-CN-NBOH has a larger binding pocket than LSD, resulting in several helices shifting outward. 25-CN-NBOH produces tighter contact and occupancy, inducing a conformational change in the key switch PIF motif, as well as other key GPCR motifs, E/DRY, NPxxY, and toggle switch, which transform to active conformation [213]. In terms of secondary structure, TM6’s outward movement allows for Gq embedding, and hydrophobic residues at the bottom of TM6 interact with Gq. The most critical contact comes from ICL2, which, in the inactive conformation, presents a rigid helical turn after stabilizing Gq. This result is consistent with a large-scale MDS study of GPCRs and G proteins: GPCRs have a conserved sequence that binds to G proteins at the bottom, as well as variable specificity recognition sequences, resulting in different subtypes of G protein selectivity [214].

To further understand the structure at various activated stages, the structures of the 5-HT_2B_ receptor linking to LSD in the transducer-free, Gq-coupled, and β-arrestin-coupled states were determined simultaneously (Figure 3B) [97]. The structure coupled to β-arrestin showed greater TM6 outward movement and strong hydrogen bond contacts whose binding to Gq was disrupted. The key motifs were in an intermediate state between transducer-free and Gq-coupled. This demonstrates that when binding to different downstream transducers, GPCRs can flexibly undergo conformational changes to adapt their embedding. If ligands are designed to selectively recruit transducers, they will effectively reduce the undesired side effects of broad recruitment [11,215].

### 5.3. Removal of Hallucinogenic Effects

Unlike LSD, psilocin adopts a unique binding pose in the 5-HT_2A_ receptor, with the lipid monoolein occupying the side-extended pocket (SEP), resulting in psilocin binding to the extended binding pocket (EBP) but not to the orthosteric binding pocket (OBP) [25]. This binding mode plays a critical role in the recruitment of β-arrestin. Mice with a key binding residue mutation in the EBP for LSD do not experience hallucinations; instead, they exhibit an antidepressant response. This suggests that the hallucinogenic effects result from the efficient co-activation of both G protein and β-arrestin. It is speculated that a relatively less efficient β-arrestin bias activation will produce antidepressant activity while avoiding hallucinogenic effects. By not occupying the lipid insertion site, a β-arrestin-biased 5-HT_2A_ receptor agonist, IHCH-7086, was designed (Figure 3C), which has both antidepressant effects and does not induce hallucinations even at high doses. Although examples are rare, a single isomerization site was confirmed in the TM3-5 intracellular region, which can accommodate lipid molecules like monoolein [216,217]. The role of membrane lipids in GPCR conformational changes needs further exploration [218].

Besides modifying drugs based on known binding modes, virtual screening is another effective strategy [9,150,219]. However, the lack of high-resolution structures and extensive databases has limited the utility of prior screenings for 5-HT_2_ receptors [220,221]. The scaffold THP, which exists in natural derivatives such as LSD and anti-migraine drugs, is underrepresented in commercial drug libraries. To address this issue, a virtual library containing 75 million THP scaffold molecules was constructed using on-demand synthesis methods [117]. Two high-affinity nitrogen heterocyclic analogues, R-69 and R-70, were discovered through large-scale virtual screening. The docking results of both molecules, after MDS optimization, closely align with cryo-EM findings. The R-69-induced conformational changes by W6.48 and PIF resemble those of LSD, and R-69 binding to residues deep within the OBP occurs. R-69 exhibits antidepressant effects that are equivalent to SSRI at a 1/40th of the dose. Furthermore, R-69 does not contact W3.28, which is believed to mediate hallucination and β-arrestin recruitment (Figure 3C). This leads to a distinct signaling bias for R-69 compared to IHCH-7086, as R-69 prefers to activate Gq while recruiting β-arrestin less efficiently.

These two studies employed drug modification and drug screening strategies to design non-hallucinogenic antidepressant compounds. A balanced model was proposed to explain how a molecule that activates the 5-HT_2A_ receptor produces hallucinogenic or antidepressant effects, or both. LSD and psilocin cause high-efficiency co-activation of G protein and β-arrestin, resulting in simultaneous hallucinogenic and antidepressant effects. However, when highly selective activation of one pathway is achieved, such as the β-arrestin-specific activator IHCH-7086 or the Gq activator R-69, the 5-HT_2A_ receptor is no longer overloaded, and its hallucinogenic effects are stripped away [25,165]. This theory will help re-examine the design of GPCR-targeting drugs, and GPCRs should be viewed as conformational microprocessors rather than simple on–off switchers.

## 6. Designing ADs for the 5-HT_1A_ Receptor

### 6.1. Structure of the 5-HT_1A_ Receptor 

Three structures of the 5-HT_1A_ receptor were released simultaneously, including the apo structure, the 5-HT-bound structure, and the aripiprazole (antipsychotic drug)-bound structure [222]. The structural mechanism of lipid-regulated activation of the 5-HT_1A_ receptor and the high selectivity for activation of apripiprazole are revealed. The distinctive feature that is shared by the three structures is that Ptdlns4P plays a vital role in stabilizing the active conformation of the 5-HT_1A_ receptor, which is the precursor of phosphatidylinositol and a key synthesis mediator of the second messenger diacyl glycerol (DG) [223,224,225]. Ptdlns4P has been identified as a positive allosteric modulator that mediates Gi recruitment. It is worth noting that cholesterol in the surrounding area inserts itself into the crack between TM1 and TM7 and directly participates in shaping the ligand pocket. This results in the 10–1000 times higher affinity of aripiprazole for the 5-HT_1A_ receptor than for other subtypes, which matches the role of cholesterol in the functional regulation of the 5-HT_1A_ receptor [226,227]. Furthermore, the 5-HT_1A_ receptor exhibits high basal activity in a physiological state due to the structured water molecules in the ligand-binding pocket, mimicking 5-HT in activating the receptor [228].

This study reveals the important roles of lipids and water molecules in regulating the 5-HT_1A_ receptor, including their roles as allosteric modulators, agonist mimics, and pocket shapers. This feature, which is not observed in other 5-HT_1_ receptor subtypes, underscores the significance and functional diversity of the 5-HT_1A_ receptor [222,229]. While the utilization of this structure remains infrequent, it can be speculated that the conformational flexibility of the 5-HT_1A_ receptor allows for diverse and even opposite functions in varying membrane and cellular environments [230].

### 6.2. Structure of the Aripiprazole-5-HT_2A_ Receptor

Another antidepressant 5-HT_1A_ receptor agonist, IHCH7041, was designed based on the unique binding mode of the third-generation antipsychotic drug aripiprazole to the 5-HT_2A_ receptor. Aripiprazole has less significant side effects compared to traditional antipsychotic drugs, which is believed to be due to its activation of the D2 dopamine receptor (DRD2) and antagonism/partial activation of the 5-HT_2A_ receptor [231,232]. In the resolved structure, aripiprazole is inserted into the binding pocket in an unexpected inverted helical posture [24,233]. The spatial hindrance by its secondary pharmacophore resulted in low affinity for the 5-HT_2A_ receptor. This inspired the development of IHCH7041, which is derived from a larger nitrogen-containing ring replacing the primary molecule of aripiprazole. IHCH7041 does not bind to the 5-HT_2A_ receptor and exhibits highly selective activation of the DRD2, improving the symptoms of schizophrenia, cognitive impairment, and depression. Its antidepressant effect has been demonstrated to come from the activation of the 5-HT_1A_ receptor. This study indirectly proves that specific activation of the 5-HT_1A_ receptor will produce antidepressant effects independently of the 5-HT_2A_ receptor.

### 6.3. Brain Region Specificity of the 5-HT_1A_ Receptor

As the earliest confirmed member of the 5-HT_1_ receptor family, the 5-HT_1A_ receptor plays an important role in the 5-HT system, regulating emotion, cognition, motor coordination, and other areas. It is distributed in multiple brain regions such as the dorsal raphe DRN, HPC, Amyg, HT, and basal ganglia [73,234]. The 5-HT_1A_ receptor at the postsynaptic site in the HPC and cortex has antidepressant effects, while the 5-HT_1A_ autoreceptor at the presynaptic site in the DRN inhibits signal transmission and 5-HT synthesis after activation, which amount to negative feedback [235,236,237]. The delayed onset of action of SSRIs is attributed to the reciprocal modulation of synaptic pre- and postsynaptic 5-HT_1A_ receptors during the initial stages of treatment. After several weeks to months of administration, desensitization of presynaptic 5-HT_1A_ receptors occurs, allowing the antidepressant effects of postsynaptic 5-HT_1A_ receptors to become evident [238,239]. Differences in 5-HT_1A_ receptor’s affinity for drug molecules in various brain regions may result from the binding pocket reshaping, which is mediated by transducers. For instance, 5-HT_1A_ receptors predominantly couple with Gαo in the HPC and with Gαi in the DRN [240,241,242]. Therefore, designing tissue-specific molecules that target only the postsynaptic 5-HT_1A_ receptor may have fast antidepressant effects.

Although long-chain aromatic piperazines, represented by buspirone, have been developed as 5-HT_1A_ receptor agonists, their off-target effects and unfavorable metabolic properties have limited their application [243,244]. The budding of highly selective 5-HT_1A_ receptor agonists came from the charge characteristics that were provided by the Weinstein model, which developed a lead structure containing a benzoylpiperidine fragment [167,245]. Further development of NLX-101, which preferentially activates postsynaptic receptors in the cortex and HPC, was supported by in vivo imaging and magnetic resonance imaging [246,247,248]. NLX-101 leans toward the ERK1/2 pathway and has therapeutic effects on Rett syndrome and antidepressant effects. Based on the docking results of NLX-101 and the 5-HT_1A_ receptor, a series of highly specific agonists that are biased towards pERK and β-arrestin were designed, revealing structural–functional selectivity features of docking poses [168,249]. Among them, NLX-204 has the highest pERK bias and shows a rapid antidepressant effect that is similar to that of ketamine in animal experiments. This suggests the feasibility of designing antidepressant molecules targeting specific subtypes of the complex 5-HT_1A_ receptor in different brain regions. Although the precise structure of the 5-HT_1A_ receptor was not resolved until recent, homology modeling has been aiding drug design since as early as 20 years ago [167,222].

In addition to designing postsynaptic 5-HT_1A_ receptor agonists, designing drugs to regulate presynaptic 5-HT_1A_ autoreceptors in the DRN is also feasible. Neuronal nitric oxide synthase (nNOS) has been found to be highly expressed in the DRN of chronically stressed depressed mice. nNOS forms a complex with SERT, reducing SERT membrane translocation during depression [250]. This leads to decreased activation of the postsynaptic 5-HT_1A_ receptor and inhibitory signals, including reduced neuron firing frequency and diminished 5-HT release in projections to the cortex, ultimately resulting in low postsynaptic 5-HT_1A_ receptor activation [251,252].

Researchers designed a complex, ZZL-7, targeting the nNOS-SERT binding site based on structural information (Figure 4B). ZZL-7 binds tightly to the groove of the nNOS PDZ domain and selectively dissociates the two proteins [253,254]. This facilitates the normal reuptake of presynaptic 5-HT and reduces the activation of 5-HT_1A_ autoreceptors. The molecule has a quick onset of antidepressant effects of only 2 h. Since the SERT-nNOS expression in the DRN is much higher than in other tissues, ZZL-7 has minimal off-target effects. 5-HT receptors are widely distributed in various brain regions and produce different physiological effects. Selecting intracellular couples that are highly expressed in specific brain regions, such as nNOS, as targets can effectively prevent undesired receptor activation. Additionally, molecules targeting protein complex binding sites have better bioactivity and cleaner structures, making them highly promising in next-generation drug development. The mechanism of action of and main information on the novel antidepressant molecules targeting the 5HT system that are mentioned above are provided in Figure 4 and Table 3.

## 7. Ketamine: Ca^2+^ Influx and Synaptic Plasticity

### 7.1. NMDAR-Centered Glutamate Hypothesis

Building on the premise that ketamine blocks NMDAR Ca^2+^ influx, various depression hypotheses related to the glutamate system have been proposed [60,256]. Ketamine’s rapid antidepressant effect is thought to result from its antagonism to Ca^2+^ influx, mediated by NMDARs in excitatory neurons. This seemingly contradictory phenomenon has led to the proposal of various glutamate system hypotheses involving multiple brain regions and different subsynaptic membrane receptors. One of the mainstream views suggests that ketamine preferentially antagonizes NMDARs in inhibitory interneurons in the midbrain, canceling their depolarizing excitation and leading to a decrease in gamma-aminobutyric acid (GABA) release [257,258,259]. This reduces the activation of GABA_A_ receptors on presynaptic neurons and cancels their inhibitory effect. Glutamate neurons are thus maintained in an excited state, rescuing the damaged synapse connections of patients with depression. This hypothesis has been verified by MK-801, an NMDAR antagonist that targets inhibitory neurons and replicates the effects of ketamine [260].

From the point of view of abnormal discharges, glutamate neurons in the LHb have been found to generate unique clustered discharges, which, when antagonized, exacerbate the effects of the reward nuclei VTA and DRN [58]. Ketamine was found to completely block this discharge, producing an antidepressant effect when delivered solely to the LHb (Figure 5A). To further address the addictive properties of ketamine in the brain, T-type calcium channels (T-VSCCs), which co-activate clustered discharges with NMDARs, have been discovered and have the potential to become a target receptor to the LHb. The research team also found that the generation of clustered discharges is due to the high expression of the potassium channel Kir4.1 in glial cells, which promotes the decrease of extracellular potassium ions and hyperpolarizes neurons [57]. This study links clues across different scales, from the LHb to glial cells, ultimately resting on three ion channels and expanding the boundaries of the glutamate hypothesis. The discovered new target T-VSCCs share a similar rationale with SERT-nNOS, which was mentioned earlier. By bypassing key receptors such as the 5-HT_1A_ receptor and the NMDAR that are present in multiple brain regions and neurons, the strategy is to select their interacting partners as targets for development to achieve tissue specificity. Furthermore, this raises a question: if NMDARs in different brain regions or subcellular environments have distinct assembly modes and structural features, could these be exploited for designing compounds targeting specific receptor subgroups?

From the perspective of signaling pathways, ketamine induces different signal cascades by antagonizing the NMDAR at different subsynaptic locations (Figure 5B). The antagonism of the postsynaptic membrane NMDAR by ketamine enhances synaptic formation and neurotrophic effects by inhibiting eukaryotic elongation factor 2 (eEF2) and promoting BDNF expression and AMPAR recycling [60,260]. On the other hand, the excessive activation of the extrasynaptic NMDAR can be detrimental, as it leads to the overexcitation of postsynaptic neurons. Clinical studies have shown that the onset of MDD is highly associated with such overactivation, including the abnormal elevation of glutamate in body fluids and the functional abnormalities in glutamate-enriched neurons that have been identified in post-mortem and neuroimaging studies [261,262,263,264]. Ketamine cancels the inhibitory effect of abnormal firing on the mammalian target of rapamycin (mTORC) by antagonizing the extrasynaptic NMDAR and transmits the signal through pS056 to the nucleus [265]. Then, eIF4E is activated, leading to the overexpression of PSD-enriched proteins and the enhancement of synaptic plasticity. Ketamine’s antidepressant effects are the result of its ability to enhance synaptic plasticity through multiple signaling pathways. In this process, the key membrane receptors AMPAR and TrkB play a role that is as significant as that of the NMDAR [77].

**Figure 5 molecules-29-00964-f005:**
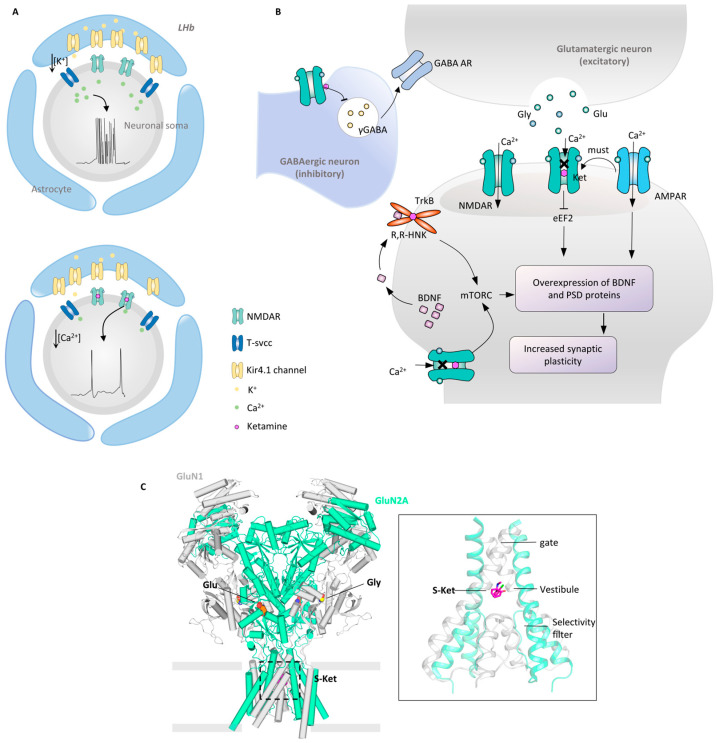
Molecular mechanism of the rapid antidepressant effect of ketamine. (**A**). Overexpression of the potassium channel Kir4.1 on the astrocyte during depression leads to a decrease in the concentration of K^+^ in the cell’s interstitial space [57,58]. Along with NMDAR and T-SVCC promoting Ca^2+^ inward flow, this leads to clustered firing of LHb neurons and oversuppression of downstream reward nuclei. In contrast, antagonism of NMDAR by ketamine abolished the abnormal cluster discharge and restored normal Ca^2+^ inward flow and action potential. (**B**). Ketamine inhibits GABA release by antagonizing NMDAR on inhibitory GABAergic interneurons, which leads to the withdrawal of its inhibitory effect on glutamatergic excitatory neurons [257,258,259]. Ketamine blocking NMDAR-mediated Ca^2+^ inflow is represented by ✕ symbol. Direct antagonism of the NMDAR on the postsynaptic membrane produces an inhibitory effect on eEF2 [60,260]. In addition, antagonism of the extrasynaptic NMDAR activates the mTORC pathway, promoting PSD-enriched protein expression and enhanced synaptic plasticity [265]. Activation of AMPAR plays a necessary role in the antagonism of NMDAR by R-HNK to produce antidepressant effects [60,260]. The transmembrane helix of TrkB can bind R-HNK and activate downstream synaptic plasticity signaling pathways via dimerization [134]. (**C**). NMDAR binding S-ketamine cryo-EM structure [26]. NMDAR is a tetramer composed of two N1 subunits (gray) and two N2 subunits (green) (left). The different composition of N2 subunits determines the type of NMDAR. One conformation of S-ketamine localized in vestibule is shown (right) [PDB: 7EU7].

### 7.2. Synaptic Plasticity: The AMPAR and TrkB

The impairment of synaptic plasticity in mPFC and HPC is a characteristic feature of MDD patients, who also exhibit neuronal atrophy in these areas, reduced synaptic density and diameter, and a decreased number and length of dendritic spines [32,266,267]. Synaptic plasticity is highly correlated with the processing and storage of information in neurons, as well as the tolerance to cross-neuronal connections and electrochemical stimulation [67]. Chronic stress-induced depression is accompanied by long-term depression (LTD) of corresponding brain regions, while the antidepressant effect of ketamine is accompanied by LTP [268]. At the molecular level, reduced reactivity of the AMAPR and the NMDAR was detected in chronic depression mouse models [267]. Receptor abnormalities that are caused by chronic stress activate the pathway of synaptic disappearance and cell apoptosis, further damaging interneuronal connections [269]. The damage to interneuronal connections causes functional abnormalities in single neurons, leading to emotional and cognitive impairment in patients, ultimately resulting in depression [270].

The thenteraction between AMPARs and TrkB, receptors related to synaptic plasticity, and ketamine has been a focus of research (Figure 5B). The more potent antidepressant effect, R-HNK, which is formed after the metabolism of ketamine, relies on the activation of the AMPAR rather than the NMDAR, and this has been confirmed using an AMPAR antagonist [132]. Furthermore, studies suggest that the antidepressant effect may arise from AMPAR activation in the synapse [271,272]. This includes NMDAR antagonists with antidepressant effects like Ro 25-6981 and the muscarinic acetylcholine receptor (mAChR) antagonist scopolamine, which increase the intracellular glutamate concentration, activating both the intracellular NMDAR and AMPAR. And a series of small molecules with antidepressant effects all depend on the activation of the AMPAR, including NMDAR partial agonists, metabotropic glutamate receptor 2 (mGluR2) and mGluR3 antagonists, and mAChR antagonists [250,272,273]. Ketamine’s primary mechanism of action in antidepressant effects is the activation of synaptic plasticity pathways, the facilitation of AMPAR expression and recycling post-internalization, and the promotion of BDNF release through AMPAR-mediated Ca^2+^ influx [251].

TrkB, a synaptic plasticity-associated receptor, is activated by the BDNF and activates the synaptic plasticity pathway through phospholipase C gamma (PLC-gama) 1, mTORC, and mitogen-activated protein kinase (MAPK). A groundbreaking study identified TrkB as a target for R-HNK and SSRIs and directly activated downstream plasticity pathways [252]. The exploration of TrkB stems from two conflicting results: R-HNK’s antidepressant effect does not depend on the NMDAR [132,134]; and the NMDAR antagonist MK-801 does not produce antidepressant effects [274,275]. This means that BDNF-TrkB signaling is necessary for ketamine’s antidepressant effects. Molecular docking and molecular dynamics simulations have shown that R-HNK can bind to its transmembrane domain and induce TrkB dimerization to activate intracellular phosphatases. This study suggests that depressive symptoms may arise from cholesterol accumulation around TrkB in lipid rafts, which hinders the formation of the necessary cross-angle between transmembrane helices, which is required for the activation of the intracellular kinase domains [276,277]. aDs can act as anchors to stabilize the cross-angle. The transmembrane dimer helix of the tyrosine kinase receptor may become a new target for AD design.

Ketamine antidepressant hypotheses are difficult to distinguish, because their causal explanations or even contradictory explanations exist due to complex interactions between signaling pathways. It is imperative to further explore whether antagonizing the Ca^2+^ influx or enhancing synaptic plasticity is the key factor which is essential for the rapid onset of ketamine’s effects, the swift reduction in suicidal ideation, and its successful application in TRD [278]. In future research, the promising approach of targeting other proteins within the synaptic plasticity pathway to design fast-acting antidepressant molecules should be investigated.

### 7.3. Structural Mechanism of the S-Ketamine NMDAR

The first NMDAR structure that was obtained was from X-ray crystallography with the channel blocker Ro25-6981, and the structures of triheteromeric were resolved by cryo-EM [279,280]. However, the mechanism of psychoactive drugs binding to NMDARs remains to be determined due to the low-resolution electron microscopy density at the time. The publication of the NMDAR coupled with ketamine structures marked a significant advancement (Figure 5C) [26]. There subtypes are GluN1-GluN2A and GluN1-GluN2B, which are enriched in the brain cortex and HPC. As a tetrameric ion channel receptor, the NMDAR has several intertwined helices in the transmembrane region. Its channel comprises ion channel gates, vestibules, and selective filters from top to bottom. S-ketamine binds in the center of the vestibule and is highly dynamic, adopting two main conformations, “biased up” and “biased down”, as discovered by MDS. R-ketamine is predicted to bind more tightly to NMDARs and produce more potent antidepressant effects. HNK’s affinity for NMDARs is notably diminished, because the hydroxyl group on its hexane chain disrupts hydrophobic interactions. This observation aligns with previous research findings that R-HNK exerts its effectiveness through the AMPAR and TrkB pathways [132,134].

Another study published the structures of NMDARs that are bound to S-ketamine, the antipsychotic drug haloperidol, and the Alzheimer’s drug memantine [79]. A higher resolution of the electron microscopy density of S-ketamine was achieved, confirming three stable conformations through molecular dynamics simulations. This is unusual, as the other two drugs exhibit only one conformation. The drugs bound to residues in the threonine (Thr), hydrophobic, and asparagine (Asn) rings, arranged from top to bottom. Three ligands were found to hinder Ca^2+^ influx by physically blocking and interacting with specific residues. As a common characteristic of channel blockers, promoting hydrogen bonding between the Thr ring and the hydrophobic ring leads to gate closure. The speed of channel closure is related to the hydrophobic interactions between the three ligands and the Thr and hydrophobic rings. The broad hydrophobic interactions of phencyclidine promote rapid channel closure, inducing severe psychosis, while memantine, with its slow channel closure, has fewer side effects. Ketamine falls between these two in terms of its effects [281,282]. The correlation between this structure and physiological effects will help design NMDAR antagonists with fewer side effects.

Can the drug design strategy for GPCRs be adapted to develop functionally selective drugs for distinct NMDAR subtypes? This relates to variations in NMDAR assembly across brain regions and subcellular compartments. To further understand the structural basis for the functional diversity of different subtypes, cryo-EM images of GluN1-N2D (located in the thalamus and HT), GluN1-N2C (located in the cerebellum and olfactory bulb), and GluN1-N2A-N2C (located in the cerebellum) and enriched in GABAergic interneurons were obtained [283]. The ion channel opening probability of the GluN1-N2D receptor is 50 times lower than that of the GluN1-N2A receptor due to the more closed N-terminal domain of the GluN1-N2D receptor, which results in its lower channel opening probability [283]. In addition, the GluN1-N2C receptor adopts a special asymmetric conformation that is different from that of the classical NMDAR. Finally, the N2A and N2C subunits in the N1-N2A-N2C tri-receptor display a conformation that is close to one protomer in the corresponding di-receptors. This study establishes the link between channel activity and NMDAR subtype structure, aiding the development of subtype-selective NMDAR probes or ADs.

A specific subtype-targeting antidepressant molecule has been discovered, YY-23, a thioglycoside from Rhizobium, which produces metabolites in the brains of mice [284,285]. YY-23 can effectively relieve stress and selectively and reversibly inhibit NMDAR-mediated currents. It can reverse various depressive symptoms, including reduced social interaction, and is faster acting than fluoxetine. A whole-cell voltage clamp confirmed YY-23 as a selective allosteric inhibitor of the GluN1-GluN2D subtype, which acts to suppress GABAergic neurons. Its inhibitory effect comes from binding to the S2 segment of the ligand-binding domain of the GluN2D subunit. The discovery of YY-23 demonstrates the potential for designing functionally selective antidepressant compounds targeting specific NMDAR subtypes. The composition, distribution, physiological effects, and corresponding available compounds of different subtypes of NMDARs have been summarized [286].

### 7.4. Ketamine Targets Multiple Types of Receptors

Ketamine has been documented to engage with diverse receptors and ion channels, encompassing those associated with opioid, cholinergic signaling, and hyperpolarization-activated cyclic nucleotide-gated (HCN) channels. Ketamine tends to have a lower affinity for these receptors than for the NMDAR, but the role of these receptors in the antidepressant effect cannot be ignored.

HCN channels represent voltage-gated cation channels [287]. The anesthetic efficacy of ketamine stems from its interaction with HCN1-HCN2 heteromeric channels and the modulation of hyperpolarization-activated pacemaker currents [288]. Notably, the administration of ketamine in mice lacking the HCN1 gene failed to ameliorate depressive behavior [289]. However, the necessary role of HCN1 in ketamine antidepressants cannot be confirmed, because the behavioral changes that are caused by HCN1 deficiency cannot be ruled out.

Ketamine has the capacity to bind to muscarinic acetylcholine receptors (mAChRs) and nicotinic acetylcholine receptors (nAChRs) [290]. mAChRs are GPCRs, and ketamine exhibits binding affinity to subtypes M1, M2, and M3 of mAChRs. nAChRs function as non-selective cation channels that are activated by the neurotransmitter acetylcholine. These receptors consist of five subunits, comprising 10 α (α1-α10) and 4 β (β1-β4) nAChR subunits [291]. Diverse combinations of these subunits give rise to numerous functional nAChR subtypes. It has been reported that ketamine serves as a non-competitive open channel blocker for the α7, α4β2, α4β4, and α3β4 nAChR subtypes [292,293,294,295,296]. The antagonistic activity of ketamine metabolites against α7 nAChR may be associated with its antidepressant effects, a phenomenon that is validated in animal models [297]. Consequently, nAChR antagonists have been investigated in clinical trials for the treatment of depression [298].

Opioid receptors, a type of GPCRs, are subdivided into three main subtypes (μ, δ, and κ). Among these, μ and κ-opioid receptors (MORs and KORs, respectively) have been identified as direct targets for ketamine, acting as partial agonists for MORs and KORs [299]. Notably, while morphine-induced activation of opioid receptors failed to elicit alterations in depressive behavior in mice [300], the administration of diverse opioid receptor antagonists, including naltrexone (which preferentially targets MORs and KORs), CTAP (a selective MOR antagonist), or LY2444296 (a KOR-specific antagonist), successfully attenuated the behavioral effects of ketamine in rodents and its antidepressant efficacy in patients with MDD [301,302,303]. Consequently, the precise contribution of opioid receptors to the antidepressant effects that are induced by ketamine warrants further elucidation.

In summary, the antidepressant effect of ketamine and its metabolites comes from the binding of their various receptors to ion channels, as well as the interaction between receptors. They are ultimately identified as key targets for antidepressant effects to design novel, safer ADs.

## 8. Conclusions

Depression imposes a huge burden on humanity, driving unceasing exploration of the pathophysiological mechanisms of depression and discovery of ADs. The development of ADs can be divided into three stages (Figure 6): (1) Before 2000, transport inhibitors targeting the monoamine system dominated the development of Ads, based on the monoamine hypothesis. However, these ADs had unexpected side effects and undesirable properties such as slow onset and low efficacy. (2) From 2000 to 2016, two landmark clinical studies re-evaluated the pathogenesis of depression by rediscovering psychedelic drugs and ketamine, which had been neglected for 50 years. New hypotheses about depression such as the glutamate hypothesis and synaptic plasticity were widely discussed. (3) From 2017 to the present, the structures of 5-HT receptors related to depression have been successively resolved by cryo-EM, and the active structures of more than 20 5-HT receptors have been published. The explosion of structural information and the expansion of the super large virtual chemical space have enabled researchers to design or screen high-receptor-selective ADs such as the IHCH-7041, ZZL-7 R-69, and IHCH7086.

The revolution in structural biology that was brought about by cryo-EM has driven the design of targeted ADs, and ideal molecules with receptor subtypes, spatial localization, and activation pattern specificity will be targeted for the next stage of research. Although most of the structures of active GPCRs have been obtained, there are still only few cases of utilizing them. By designing antidepressant molecules that are directed to activate G protein isoforms and β-arrsetin and target different subtypes of the NMDAR, a combination of rapid onset of action and low-level side effects ADs will be achieved. This cannot be achieved without VDS advances, including the expansion of the virtual chemical space, the improvement in molecular docking efficiency, and the fine construction of MDS environments. At the same time, these breakthroughs will drive industrial change, and structural modeling will serve as the starting point for drug discovery, replacing the traditional high-cost, high-side-effect drug screening methods. We are standing at the starting point of the next-generation drug development, which is a new era of rational drug development with the rational development of drugs from the billion-level drug space and atomic-level resolution, and novel ADs are undoubtedly the most representative examples.

## Figures and Tables

**Figure 1 molecules-29-00964-f001:**
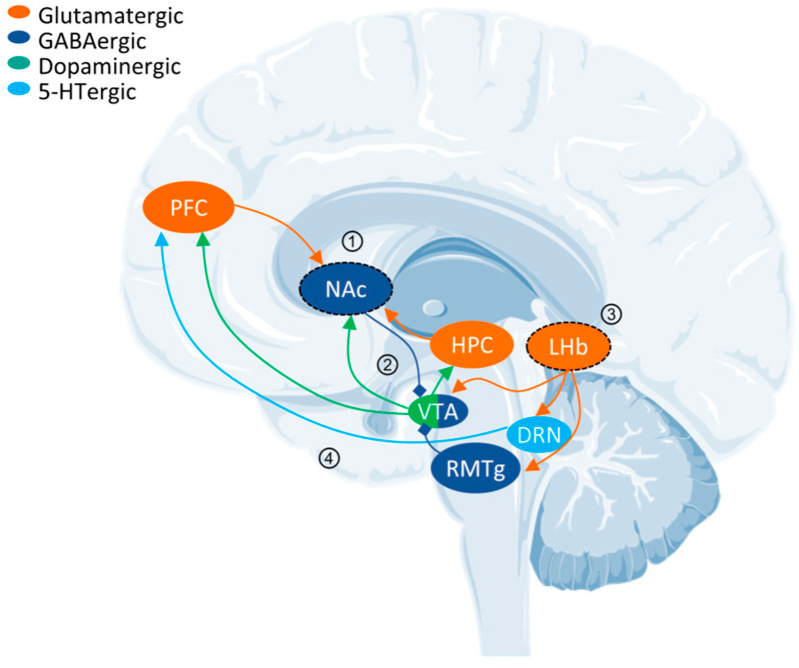
Key neural circuits in MDD. Reward circuitry: NAc integrates excitatory neurotransmission from HPC and mPFC to regulate emotions (process 1). The VTA-NAc dopaminergic pathway displays antidepressant-like properties (process 2). LHb is linked to aversion and depressive states, stimulating RMTg, which inhibits VTA dopaminergic neurons. LHb also directly suppresses the reward centers VTA and DRN (process 3). 5-HTrgic neurons project from DRN to mPFC: the activation of 5-HT_1A_ autoreceptors in the DRN reduce serotonergic neuron activity, leading to decreased 5-HT release in the mPFC (process 4). The types of neurons in circuits are distinguished by the color of the arrows and shown at the top left. The arrows of excitatory projection are triangular, and inhibitory projection are prismatic. Abbreviations: PFC, prefrontal cortex; NAc, nucleus accumbens; HPC, hippocampus; VTA, ventral tegmental area; RMTg, rostromedial tegmental nucleus; LHb, lateral habenula; DRN, dorsal raphe nucleus.

**Figure 2 molecules-29-00964-f002:**
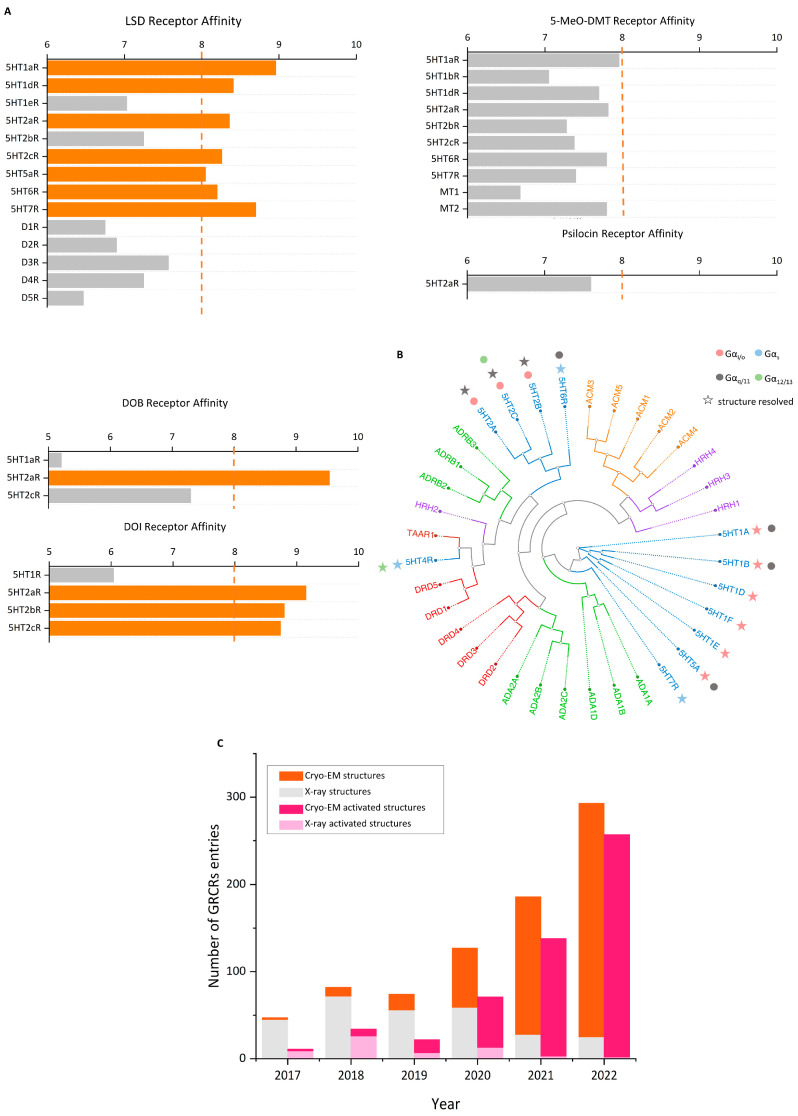
Affinity and structural results of psychedelics for GPCRs. (**A**). Three chemical types of psychedelics with different receptor binding affinities. LSD broadly activates multiple 5-HT receptor subtypes with high affinity (receptor affinity > 8.0, displayed in orange) and dopamine receptors (DRs). The two indoleamines have a low affinity, and the two phenylalkylamines have a high affinity for 5-HT_2_ receptors. Data from ChEMBL (https://www.ebi.ac.uk/chembl, accessed on 1 November 2023), receptor affinity = pKi. Abbreviations: LSD: lysergic acid diethylamide, 5-MeO-DMT: 5-methoxy-N,N-dimethyltryptamine, DOI: 2,5-Dimethoxy-4-iodoamphetamine, DOB: 4-Bromo-2,5-dimethoxyamphetamine. (**B**). Phylogenetic tree of aminergic GPCRs, GPCR cluster based on sequence similarity. The 5-HT receptors–coupled G protein subtypes are displayed as dots, and the combinations that have been resolved are represented as stars. (**C**). Cryo-EM and X-ray resolved growth trends of all and activated GPCRs (2017–2022). (**B**,**C**) data from GPCR DB (https://gproteindb.org, accessed on 1 November 2023).

**Figure 3 molecules-29-00964-f003:**
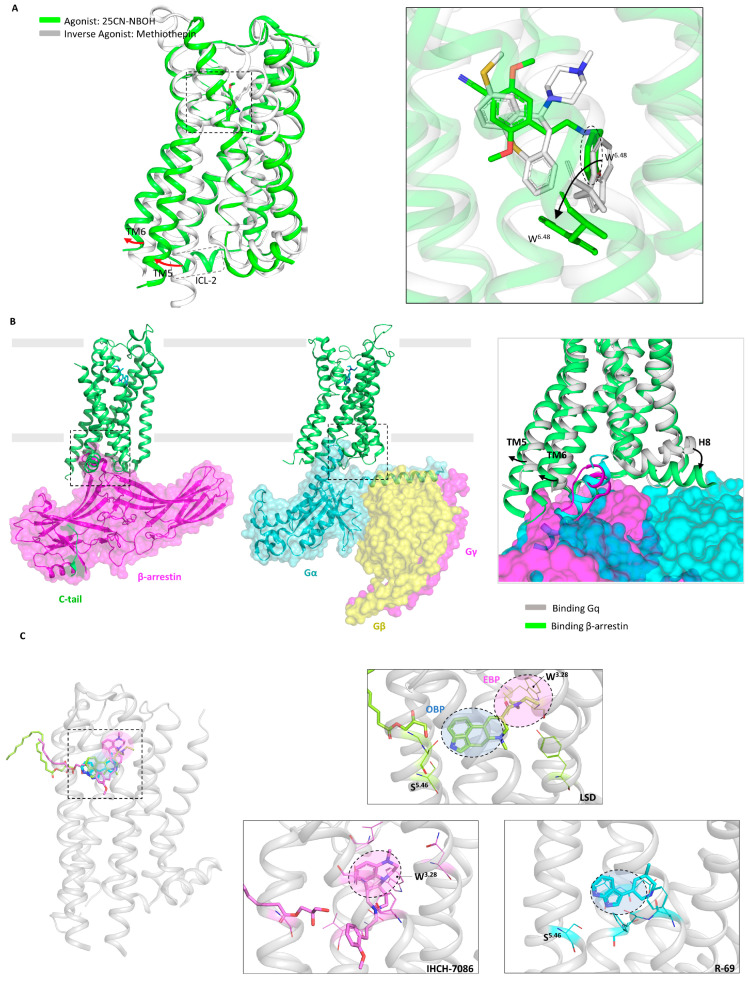
Structure–function mechanism of psychedelic activation of 5-HT_2A_ receptor. (**A**). 5-HT_2A_ receptor binds an agonist, 25CN-NBOH [green, PDB: 6WHA], and an inverse agonist, methiothepin [gray, PDB: 6WH4], exhibiting active and inactive conformations, respectively [24]. The active conformation of 5-HT_2A_ receptors TM5 and TM6, moving outward, and ICL2, transitioning from a free loop to an incomplete helix. The details in the dashed line box are shown on the right, the same below. (**B**). 5-HT_2B_ receptor snapshots of linked β-arrestin [green, PDB: 7SRS] and Gq [gray, PDB: 7SRR] [97]. Embedding Gq and β-arrestin relies on the outward mobility of TM5 and TM6. β-arrestin, leads to more significant outward movement, accompanied by a downward shift of helix 8 (H8). (**C**). LSD [yellow, PDB: 7WC6] occupies both the OBP and EBP, contacting key residues S5.64 and W3.28 in both pockets, respectively. IHCH-7086 [red, PDB: 7WC9] is mainly located in the SEP pocket, while the other pharmacophore avoids the OBP, leaving room for lipid occupancy [25,165]. Most of the body of R-69 [blue, PDB: 7RAN] is in the OBP and does not contact W3.28.

**Figure 4 molecules-29-00964-f004:**
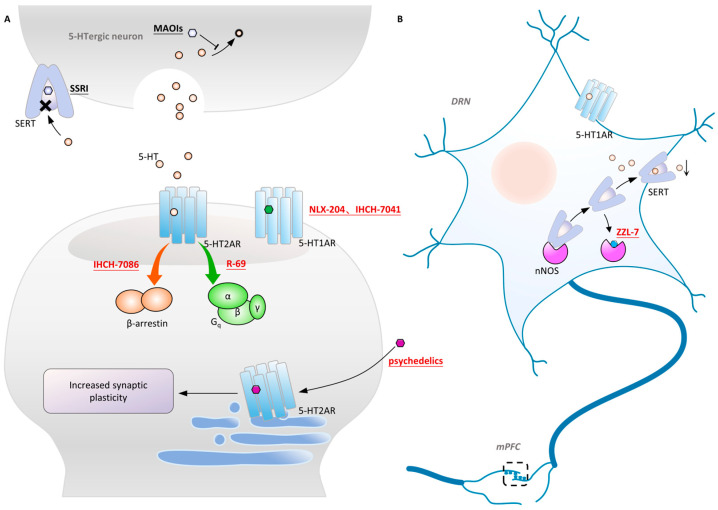
Schematic diagram of antidepressant molecules acting on the 5-HT system. (**A**). From left to right, they are as follows: SSRI increases 5-HT concentration in the synaptic gap by antagonizing. SERT blocks the reuptake of 5-HT into the presynaptic membrane. MAOIs reduce the deactivation of 5-HT after oxidation reaction by inhibiting MAO activity. IHCH-7086 recruits β-arrestin through functionally selective and mild activation of 5-HT_2A_ receptors, thereby removing hallucinogenic activity [25]. R-69 activates 5-HT_2A_ receptor with high affinity and specificity, with a bias toward activation of Gq [165]. IHCH-7041 selectively binds DRD2 and 5-HT_1A_ receptor but not 5-HT_2A_ receptor. [233]. NLX-204 highly selectively activates the postsynaptic 5-HT_1A_ receptor of mPFC without binding to the 5-HT_1A_ autoreceptor of DRN [249]. Psychedelics activate intracellular (especially endoplasmic reticulum) 5-HT_2A_ receptor and enhance synaptic plasticity [71]. (**B**). ZZL-7 facilitates the translocation of SERT to the plasma membrane by disrupting the linkage between nNOS and SERT [253]. This leads to a decrease in extracellular 5-HT concentration. The 5-HT_1A_ autoreceptor activation located in the DRN is inhibited, thereby abolishing the inhibition of the mPFC projection impulse.

**Figure 6 molecules-29-00964-f006:**
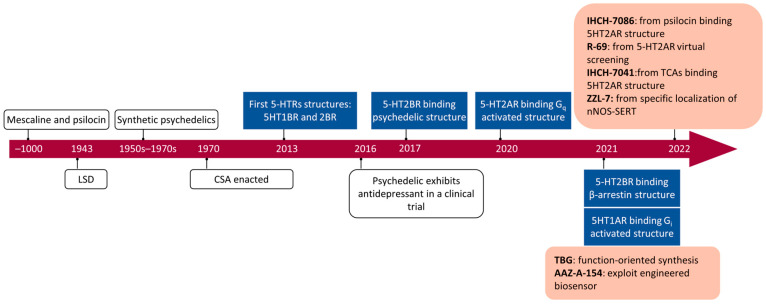
Landmark events in the discovery and utilization of psychedelics. The representative psychedelics are found in the white box. The structural discoveries are shown in blue boxes. The development of novel antidepressant molecules is shown in pink boxes.

**Table 2 molecules-29-00964-t002:** Large virtual screening tools for GPCRs and their websites.

Name	Website *	Introduction	Reference
Virtual drug libraries			
ZICN 15/20/22	https://zinc15.docking.org, https://zinc20.docking.org https://cartblanche22.docking.org/	Zinc 15/20 contains over 980 million compounds, of which 230 million are available for purchase. ZINC-22 focuses on make-on-demand compounds and has about 37 billion molecules in 2D and 4.5 billion in 3D.	[149,158,188]
ChEMBL	https://www.ebi.ac.uk/chembl/	ChEMBL is a manually curated database of bioactive molecules with drug-like properties. It brings together chemical properties and bioactivity and includes 2.4 million compounds and 1.5 million assays.	[189]
Drugbank	https://go.drugbank.com/	DrugBank is a web resource containing detailed drug, drug target, drug action, and drug interaction information about FDA-approved drugs.	[190]
Protein structure databases			
EMDB	https://www.ebi.ac.uk/emdb/	EMDB is a public repository for electron cryo-microscopy volume maps and tomograms of macromolecular complexes and subcellular structures, which contains more than 26,000 entries.	[191]
RCSB PDB	https://www.rcsb.org/	RCSB PDB is an archive of 3D structure data for large biological molecules (proteins, DNA, and RNA). It contains more than 203,863 experimental structures and 1,068,577 computed structure models.	[116]
GPCRdb	https://gpcrdb.org/	GPCRdb contains all human non-olfactory GPCRs (and >27,000 orthologs), G-proteins, and arrestins. It includes drugs, in-trial agents, and ligands, with activity and availability data. GPCRdb annotates all published GPCR structures and provides structure models.	[145]
Protein structure prediction programs			
Alphafold2 (v2.3.0)	https://alphafold.com/ (database) https://github.com/deepmind/alphafold (program)	AlphaFold utilizes a machine learning method, enabling prediction of a protein’s 3D structure from its sequence. The database has released 200 million protein structure predictions, covering virtually all proteins.	[192]
Rosettafold	https://github.com/RosettaCommons/RoseTTAFold	Rosettafold accurately predicts protein structures and interactions using a 3-track neural network. The simultaneous processing of sequence, distance, and coordinate information by the three-track architecture assists with incorporating constraints and experimental information.	[193]
GPCRdb	https://gpcrdb.org/	GPCRdb contains all human non-olfactory GPCRs (and >27,000 orthologs), G-proteins, and arrestins. It includes drugs, in-trial agents, and ligands, with activity and availability data. GPCRdb annotates all published GPCR structures and provides structure models.	[145]
Molecular docking tools			
Dock (3.6)	https://dock.compbio.ucsf.edu/DOCK3.6/	The DOCK algorithm addresses rigid body docking using a geometric matching algorithm to superimpose the ligand onto a negative image of the binding pocket. It is suitable for tackling large library screens.	[194]
Schrödinger Glide (2023-4)	https://www.schrodinger.com/products/glide	Glide is a commercial docking software from Schrödinger. It can perform flexible ligand docking. Glide offers multiple speed vs. accuracy options for scoring modes.	[195]
VirtualFlow	https://virtual-flow.org/	VirtualFlow, a highly automated and versatile open-source platform, scales linearly with the number of CPUs that can prepare and efficiently screen ultra-large libraries of compounds.	[159]
V-SYNTHES	https://github.com/katritchlab/V-SYNTHES	A modular synthon-based approach—V-SYNTHES—for performing hierarchical screening. V-SYNTHES identifies the best scaffold–synthon combinations as seeds and iteratively elaborates these seeds to select complete molecules.	[160]

* All websites in stable accessed on 1 November 2023.

**Table 3 molecules-29-00964-t003:** Potential antidepressant molecules targeted by 5-HT system.

Name andStructure	Prototype	Target andBiased Selectivity	Discovery Method	Pharmacology	References
TBG 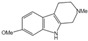	Ibogaine 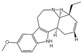	High selectivity for 5-HT_2_ receptors and weak or no opioid agonist activity.	Applying the principles of function-oriented synthesis.	Zebrafish toxicity assay: low cardiotoxicity, low lethality.HTR assays: not hallucinogenic.Transcranial 2-photon imaging: increased spine formation.Forced swim test behavior: significantly reduced immobility.Alcohol- and Heroin-seeking behavior: reduced both intakes.	[196]
AAZ-A-154 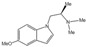	DMT 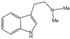	Binds 5-HT_2A_R but not the hallucinogenic conformation.	Using psychLight2-expressing cell line imaging screening platform.	HTR assays: not hallucinogenic.Forced swim test: rapid and long-lasting antidepressant-like effects after a single administration.Sucrose preference: reduced anhedonia in depressive mice for at least 12 days.	[197]
IHCH-7086 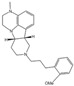	Lumateperone 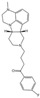	5-HT_2A_R β-arrestin-biased.	Based on the position of Psilocin, identifying crucial residues for β-arrestin.	HTR assays: failed to produce any HTR, even at high doses.Forced swim test and tail suspension test: significantly attenuated acute restraint stress-induced/corticosterone-induced depression-like behavior.	[25]
R-69 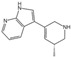	THP 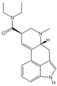	Highly selective activation of 5-HT_2A_R and stimulated Gq signaling.	Docking a designed 75 million THP scaffold library against a 5-HT_2A_R model.	HTR assays: induced very low levels of HTRs, blocked the HTRs induced by LSD.Open field locomotion: did not possess locomotor-stimulating or reinforcing activity.Forced swim test and tail suspension test: antidepressant-like actions at least for 24 h.Sucrose preference: substantially increased.	[165]
NLX-204 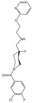	NLX-101 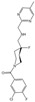	A 5-HT_1A_R ERK-biased agonist in PFC.	Characterization of NLX-101 specific activation of ERK signaling in the PFC/HPC.	ERK1/2 phosphorylation: dose-dependent activation in the rat PFC.Forced swim test: antidepressant-like effects.Sucrose preference: rapid effect, reversing sucrose consumption deficit.	[255]
ZZL-7 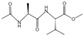	Sakura-6 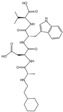	nNOS PDZ domain in DRN.	Dissociating the SERT from nNOS reduced intercellular 5-HT concentration in DRN.	In vivo electrophysiology: increased firing frequency of serotonergic neurons.Forced swim test and tail suspension test: reduced immobility time.General activity: no effect and no other side effects.	[253]
IHCH-7041 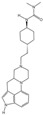	Aripiprazole 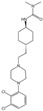	A partial agonist at DRD2/3 and 5-HT_1A_R with negligible 5-HT_2A_R binding.	Based on TGAs adopting an unexpected “upside-down” posture in the 5-HT_2A_R binding site.	Locomotor responses: displayed antipsychotic-like effects.Forced swim test and tail suspension test: Significantly attenuated immobility.Novel object recognition: significantly attenuated deficits.Morris water maze: restored their spatial navigation ability.In vivo electrophysiology: inhibited glutamatergic transmission through 5-HT_1A_R.	[233]

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
