# Peer review of "Exploring Novel Antidepressants Targeting G Protein-Coupled Receptors and Key Membrane Receptors Based on Molecular Structures"

_molecules, 2024, doi:10.3390/molecules29050964_

Round 1

Reviewer 1 Report

Comments and Suggestions for Authors

The present manuscript by Yao and co-workers is related to the actions of novel antidepressants on G protein-coupled receptors.

In this work, the authors review the sites of interaction and mechanisms of action of two categories of fast-acting antidepressants: S-ketamine and its metabolites, and psychedelics.

They focus on several aspects such as the structure of receptors intimately related to depression and their interaction with the novel antidepressants mentioned above. These receptors mainly include N-methyl-D-aspartate receptor (NMDAR) and 5-hydroxytryptamine 2A (5-HT2A) receptor. Very interestingly, the authors mention the subtypes of these receptors (with specific subunit combinations) and their subcellular location in specific brain regions (the cortex and subcortex, the hippocampus, amygdala, nucleus accumbens, medial prefrontal cortex, lateral habenula) importantly associated with major depression disorder, which may be potential targets for these novel antidepressants, having highly selective, rapid onset and low side effects.

The subject of this study is very exciting and interesting, and highly relevant with clinical repercussions. The work is excellently documented with recent findings on this topic, and clearly presented.

Specific comments for the manuscript:

Some spelling words.

Line 38, “virtual” instead of vrtual.

Line 692, “Brain” instead of Barin.

Line 776, “GABA” instead of γGABA.

From line 44 to 45, the size of the characters changed.

It is documented that ketamine interacts with other membrane proteins: nicotinic and muscarinic acetylcholine receptors, opioid μ-receptor, among others. Please include documentation of the antidepressant compounds reviewed here that target other membrane proteins (receptors, ion channels, transporters, etc).

This comment does not pretend to be a critic of the work. For this reviewer is amazing and hard to believe that given the multifactorial complexity of depression, with a high diversity of symptoms (feelings of guilt, hopelessness, psychiatric and cognitive impairments, disturbances in sleep and appetite), in which several brain regions, neurotransmitter systems, subtypes of ionotropic and metabotropic receptors and their differential subcellular location, as stated by the author, references 29-31) are implicated, that one compound such as S-ketamine, its metabolites, or psychedelics, even with their apparent highly selectivity on NMDAR and 5-HT2A receptor, rapid onset, and low side effects in depressive subjects, be sufficient for reaching healthy in depressive subjects.

As the authors state (lines 796-798): Furthermore, this raises an intriguing question: NMDARs in different brain regions or subcellular environments have distinct assembly modes and structural features, whether could be exploited for designing compounds targeting specific receptor subgroups?

Possible along the time and with more research, it will be found that these novel antidepressants target more proteins, as occurred with, for instance, fluoxetine, that at the beginning it was considered as a highly selective serotonin reuptake inhibitor; and today we know that it targets much more membrane proteins relevant for antidepressant effect and/or side effect.

Reviewer 2 Report

Comments and Suggestions for Authors

The paper titled "Exploring Novel Antidepressants: Targeting G Protein-Coupled Receptors and Key Membrane Receptors Based on Molecular Structures" provides a comprehensive exploration of the challenges inherent in developing effective antidepressants. It prominently features breakthroughs achieved in comprehending depression-related receptors, specifically the structures of NMDAR and 5-HT2A receptors, elucidated through cryo-electron microscopy (cryo-EM). Furthermore, it extensively examines the potential therapeutic value of 5-HT1A receptors, AMPAR, and TrkB as novel targets for antidepressants. The paper aims to contribute insights into designing antidepressants that ensure rapid onset and minimal side effects, leveraging structure-based approaches.

However, the length of the review might impede its accessibility and readability. It's advisable for the author to consider condensing the content to enhance its readability while maintaining the depth of information.

Several specific points need attention for a more refined manuscript:

1.     Figure 2: Clarify the significance of “binding affinity=10-lg(Ki)”. Consider using "Kd" (dissociation constant) for increased relevance in this context.

2.     Table 1: Summarize the binding structures mentioned within this table to derive more conclusive insights.

3.     Figure Label Mistake: Rectify the labeling error from Fig 2b to 2B for accuracy.

4.     Repetitive Information: Streamline or condense the information presented in Table 2 and Line 379 to avoid redundancy and maintain conciseness.

5.     Table 2: Organize the numerous basic databases into distinct groups for ease of reference and quicker comprehension.

6.     Figure 3: Specify "W3.28" in Figure 3 to represent the mean more explicitly for better understanding.

Reviewer 3 Report

Comments and Suggestions for Authors

GENERAL COMMENTS

This rather lengthy review organizes the data from a very narrow experimental viewpoint. The tone is sometimes too polemic and sometimes reads too much like a sales pitch for cryo-EM (which this reviewer is not competent to assess), annoyingly so. The authors should tone down their language.

On the other hand, there is a dire lack of information on clinical trials of the medication candidates the authors champion.

This reviewer, who started to read the manuscript with great interest about what insights into the current antidepressant drug development it might provide, became confused, learning nothing memorable beyond what she already knew.

For all these reasons, this reviewer’s recommendation is to reject the manuscript.

SPECIFIC ITEMS

Abstract

line 35: “side effects”. You mean “adverse drug reactions”, correct? If yes please change. If you still mean “side effects”, please describe them and categorize them as “potentially useful” vs “adverse”.

Introduction

line 46: To claim that antidepressants display an onset of action after months is a misleading assertion. Please correct.

line 52: Compounds do not “struggle”, researchers and clinicians do. Please correct.

line 60: What is a “chemical space”? Please explain or correct.

line 66: “which activating … (5HT) receptors” is grammatically wrong. Please correct.

line 76: “In particular, … is not a complete sentence. Please correct.

line 102: “At the systemic scale, …” What do you mean by “scale”? “perspective”? “aspect”? Please rephrase, also the term “cross-scale abnormalities” (line 88)

line 108: “Based on …” Again, this is an incomplete sentence. Please correct.

Fig 1

Legend: Please give the full names of the brain regions involved, not just the abbreviations.

Fig 2

Figure 2 is inacceptably overburdened with graphics. Only when fully extending the pdf file over an 27-inch screen (Dell U2715H) became the invidivual descriptors legible for this reviewer. With respect to its contents, figure 2 can be split into 3 figures, i.e., the present figure 2A, 2B, and 2C which currently present completely unrelated aspects of this review. As has been noted for fig 1 already, the legend should give the full names, at least for 5-MeO-DMT, DOI, and DOB. The full name of 5-MeO-DMT is not even given in the text. “10-lg(Ki)”: This reviewer knows what you, the authors, mean by that, but the mathematical notation is not correct; please also give the dimension (unit) for Ki. “highly specific affinity” is not a scientific term. Please correct.

Comments on the Quality of English Language

extensive editing required

Round 2

Reviewer 3 Report

Comments and Suggestions for Authors

Unfortunately, the manuscript was not improved at all.

@font-face {font-family:"Cambria Math"; panose-1:2 4 5 3 5 4 6 3 2 4; mso-font-charset:0; mso-generic-font-family:roman; mso-font-pitch:variable; mso-font-signature:-536870145 1107305727 0 0 415 0;}@font-face {font-family:Calibri; panose-1:2 15 5 2 2 2 4 3 2 4; mso-font-charset:0; mso-generic-font-family:swiss; mso-font-pitch:variable; mso-font-signature:-536859905 -1073732485 9 0 511 0;}p.MsoNormal, li.MsoNormal, div.MsoNormal {mso-style-unhide:no; mso-style-qformat:yes; mso-style-parent:""; margin:0cm; mso-pagination:widow-orphan; font-size:12.0pt; font-family:"Calibri",sans-serif; mso-ascii-font-family:Calibri; mso-ascii-theme-font:minor-latin; mso-fareast-font-family:Calibri; mso-fareast-theme-font:minor-latin; mso-hansi-font-family:Calibri; mso-hansi-theme-font:minor-latin; mso-bidi-font-family:"Times New Roman"; mso-bidi-theme-font:minor-bidi; mso-font-kerning:1.0pt; mso-ligatures:standardcontextual; mso-fareast-language:EN-US;}.MsoChpDefault {mso-style-type:export-only; mso-default-props:yes; mso-ascii-font-family:Calibri; mso-ascii-theme-font:minor-latin; mso-fareast-font-family:Calibri; mso-fareast-theme-font:minor-latin; mso-hansi-font-family:Calibri; mso-hansi-theme-font:minor-latin; mso-bidi-font-family:"Times New Roman"; mso-bidi-theme-font:minor-bidi; mso-fareast-language:EN-US;}div.WordSection1 {page:WordSection1;}

Comments on the Quality of English Language

Extensive editing of English language required.